# Large Language Models as Automatic Annotators and Annotation Adjudicators for Fine-Grained Opinion Analysis

## Abstract

Fine-grained opinion analysis of text provides a detailed understanding of expressed sentiments, including the addressed entity. Although this level of detail is valuable, annotating opinions in datasets for model training requires considerable human effort and substantial cost, especially across diverse domains and real-world applications. To address this shortage of domain-specific labelled datasets, we explore the feasibility of LLMs as automatic annotators for fine-grained opinion analysis. We use a declarative annotation pipeline, an approach that reduces the variability of manual prompt engineering when using LLMs to identify fine-grained opinion spans in text. We also present a dedicated methodology for an LLM to adjudicate multiple labels and produce final annotations. We trial the pipeline with models of different sizes for the Aspect Sentiment Triplet Extraction (ASTE) and Aspect-Category-Opinion-Sentiment (ACOS) analysis tasks. In this work, we attempt to develop fully autonomous LLM-based annotators, but our results reveal an uneven picture characterised by a critical performance bifurcation: LLMs are reliable at the span level yet struggle to faithfully reproduce the relational structures that connect those spans. This suggests that LLMs are better positioned as high-fidelity annotation assistants and data augmentation tools to expand fine-grained opinion-annotated datasets, rather than replacing human annotators entirely.

## 1 Introduction

There has been a considerable evolution in the field of fine-grained opinion analysis, which has led to many theoretical and practical formulations of opinions and sentiment. These fine-grained methods provide a clearer picture of not just the sentiment polarity (positive, negative, neutral) but also the contextual elements of the aforementioned sentiments. Consider the following example:

*"I had hoped for better battery life, as it had only about 2–1/2 hours doing heavy computations (8 threads using 100% of the CPU)."*

In the example above, a *negative* sentiment is expressed according to sentence- or document-level annotations. But with a fine-grained formulation, the following additional annotations are desired (i) an opinion span expressing the polarity, in this example that is *"hoped for better"*, (ii) an explicit instantiation of the target in the text called an aspect term (in the example: *"battery life"*), They together form an aspect sentiment triplet as specified according to the Aspect Sentiment Triplet Extraction (ASTE) task (Wu et al., 2020).

The annotation can include a more coarse-grained property called an aspect category. In the above example *"Battery#Operational_performance"*, the attribute expresses the entity's attribute towards which the opinion is specifically directed. Aspect categories enable encoding cases in which the aspect term is missing, and the target sentiment is expressed implicitly. In addition to triplets elements, the quadruple annotations are assigned categories too, adhering to Aspect-Category-Opinion-Sentiment (ACOS) guidelines (Cai et al., 2021).

The existing datasets used for fine-grained opinion mining tasks are derived from SemEval tasks and are relatively small, limited to specific domains, and collected exclusively from review websites. As highlighted

in (Zhang et al., 2023), these datasets are helpful in comparing predictive methods and systems, but are not sufficient for addressing real-world use cases. Subjective opinion annotation by human annotators is a non-trivial task and requires strict adherence to annotation guidelines, as documented in (Toprak et al., 2010). While human annotation remains the gold standard for opinion annotation, its lack of scalability motivates the assessment of automatic annotation approaches.

In this work, we investigate the potential of LLMs to automatically annotate text with fine-grained opinions. The LLMs have been shown to be few-shot learners (Kojima et al., 2022) and exhibit instruction following and reasoning capabilities (Hao et al., 2024). Through our study, we enquire the extent to which they can successfully annotate fine-grained opinions: *can LLMs replace human annotators for fine-grained opinion analysis, and if not, where precisely do they fall short?* Understanding the efficacy of LLM-driven annotations can help us make an informed decision on the extent to which these models can fulfil the opinion annotator role or assist human annotators.

We also propose a dedicated LLM-based Adjudication approach that combines multiple opinion annotations for a given text input to resolve inter-annotator disagreement and obtain final opinion annotations. This is inspired by the process of annotation adjudication (Hovy & Lavid, 2010), which includes making a final judgement after considering the conflicting annotations from multiple annotators. Commonly used adjudication methods, such as traditional majority voting, assume a discrete label set. However, the opinions for ASTE and ACOS are extracted as triplets and quadruples, with an open-ended, combinatorial output space, rendering traditional majority voting impractical, if not inapplicable (Uma et al., 2021), further motivating our adjudication method. Our main contributions include:

- **Declarative LLM annotation pipeline**: We propose a schema-constraint pipeline that uses LLMs declaratively, without hand-crafting prompts, thereby enabling a more systematic interaction for extracting opinions expressed with an ASTE and ACOS specification.

- **LLM as annotation adjudicator**: We introduce a dedicated method for aggregating annotations from multiple LLM-driven annotators to produce final annotations using LLM as an annotation adjudicator.

- **Assessment of LLM at different scale**: We experiment with multiple LLMs with varying scales in our pipeline. This establishes an understanding of the trade-off between scale and desired efficiency in the task of automatically annotating opinions.

- **Empirical analysis of LLM annotation reliability**: We evaluate opinion triplets and quadruples using element-wise and pairwise metrics, revealing that while LLMs reliably extract individual spans, their alignment with human annotations degrades markedly as relational complexity increases.

## 2 Related Work

**Opinion Formulations**  Opinion[1] Mining and sentiment analysis have been well explored in natural language processing. Aspect-Based Sentiment Analysis (ABSA), the foundation of fine-grained opinion mining, evolved from feature-based summarisation (Hu & Liu, 2004; Liu & Zhang, 2012), which involves extracting and summarising opinions about features (attributes/ keywords). In this paper, we experiment with two fine-grained and extractive formulations of downstream ABSA tasks: Aspect Sentiment Triple Extraction (ASTE) (Xu et al., 2020; Wu et al., 2020) and Aspect-Category-Opinion-Sentiment Quadruple (ACOS) extraction(Cai et al., 2021).

**LLMs as Annotators**  Key steps in dataset creation for building any intelligent system include data selection, guideline development, human annotation, quality estimation, and adjudication (Klie et al., 2024). With the improvement of the general ability of LLMs, they have been investigated for annotating various subjective tasks, including span annotations(Kasner et al., 2025), argument quality annotations(Mirzakhmedova

---

[1]Unless stated otherwise, we use the term opinion as a broad concept that covers sentiment and its associated information, such as opinion target and the person who holds the opinion, and use the term sentiment to mean only the underlying positive, negative, or neutral polarity implied by opinion.

et al., 2024) and propaganda span annotations(Hasanain et al., 2024). In this work, we extend this line of investigation to fine-grained opinion annotation, exploring LLMs as annotators using inference-time adaptation. Unlike supervised approaches that require task-specific fine-tuning, our method optimises prompts rather than model weights, operating in the same training-free spirit as earlier rule-based methods (Qiu et al., 2011) while leveraging the broader reasoning capabilities of modern LLMs.

**Prompt Engineering** LLMs are known to be sensitive to the prompts (Zhuo et al., 2024) and exhibit a higher level of variability in performing complex tasks (Wang & Wang, 2025). To minimise the spurious interaction between the prompt and model selection as a confounding variable, we adopt **DSPy** (Khattab et al., 2024) and the approach of *"programming LLMs"* rather than prompting them naively. It shifts the focus from performing string manipulation on prompt strings to programming with structured, declarative modules that automatically prepare prompts using a small sample of annotated examples.

## 3  Dataset

(a) ACOS & ASTE Specifications

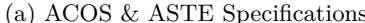

(b) Dataset Breakdown

| Task | Dataset | Train | Dev | Test |
|------|---------|-------|-----|------|
| ASTE | lap14 | 906 | 219 | 328 |
|      | res14 | 1126 | 310 | 492 |
|      | res15 | 607 | 148 | 322 |
|      | res16 | 857 | 210 | 326 |
| ACOS | laptop | 2934 | 326 | 816 |
|      | restaurant | 1530 | 171 | 583 |

Table 1: Fine-Grained Opinion Examples

| Task | Input(s) | Output |
|------|----------|--------|
| ACOS | *"I have eaten here three times and have found the quality and variety of the fish to be excellent"* | (quality, excellent, Pos, food#quality) (variety, excellent, Pos, food#style) |
| ASTE | *"moules were excellent, lobster ravioli was very salty."* | (moules, excellent, Pos) (lobster ravioli, salty, Neg) |

The datasets used in our experiments are described in Table 1b. Because fine-tuning LLMs is not an objective of our annotation pipeline, we do not utilise training splits. Instead, we use the development split to provide In-context Learning (ICL) examples, which are included in the optimised prompt. The test splits remain unchanged, allowing us to evaluate performance across multiple LLMs in our experiments. The schema in Figure 1a illustrates the relationship between the two formulations we explore in this work.

- **ASTE**: Aspect Sentiment Triple Extraction (ASTE) falls under the umbrella of ABSA tasks. Proposed by Peng et al. (2020), it is a task for extracting opinion triplets, thus providing one-shot answer to: **What** target is being discussed, **How** is the sentiment (e.g. positive) and **Why** is this sentiment (e.g. "*excellent*").

- **ACOS**: This specification Aspect-Category-Opinion-Sentiment(ACOS) Quadruple Extraction, to extract aspect, aspect category, opinion and sentiment as quadruples in text and provide full support for aspect-based sentiment analysis with implicit aspects and opinions [2]. The primary focus of this formulation is the fine-grained analysis of the opinion target. This task is more challenging than ASTE because it not only adds a category on top of the existing opinion triplet, but also takes into account implicit cases (i.e. missing explicit aspect terms or opinion spans) (Cai et al., 2021).

---

[2]Opinion in ACOS task refers to the span that expresses sentiment.

# 4 Methodology

The fine-grained opinion annotation pipeline has multiple stages: (i) Preparing and evaluating LLM-specific prompts, (ii) Redundant Opinion annotation using multiple LLMs, and (iii) Annotation adjudication using LLM to create collaborative opinion annotations.

## 4.1 Problem Definition

On a conceptual level, the process of annotating opinions using the fine-grained opinion formulations involves identifying a set of opinions expressed in the input text $T_i$. As with human annotators, the process generates multiple annotations for the same input $T_i$, followed by adjudication that resolves disagreements between annotators to produce final annotations. These steps can be formalised as:

**LLM as opinion annotators** Given the input text $T_i$ from corpora $\mathcal{T}$, our objective is to extract all expressed opinions $\mathcal{O}_i = \{o_1, o_2, ..o_n\}$. The elements of single opinion $(o_j)$ is task dependant:

- ASTE opinions are formulated as the triple $o_j = (at, s, op)$, where $at$ is the aspect term, $s$ is the sentiment and $op$ is the opinion term.

- ACOS opinions are quadruples $o_j = (at, s, op, ac)$, where $at$, $s$ and $op$ encode similar concepts as ASTE and $ac$ is aspect category made up of entity $E$ and attribute $A$ (expressed as $E\#A$).

The LLM-driven annotation pipeline maps input corpora $\mathcal{T}$ to $\mathcal{O}^k$ using $k$ LLM-based annotators.

**LLM as annotation adjudicators** We use LLM to perform the adjudication to combine opinions annotated by $k$ different LLMs to obtain the final opinion annotation, i.e.,

$$\mathcal{A}\mathrm{dj}_{\mathrm{LLM}} : (\mathcal{T}, \mathcal{O}^1, \ldots, \mathcal{O}^k) \to \mathcal{O}$$

This method, in its formulation, is similar to the stacking (Wolpert, 1992) technique in ensemble machine learning.

## 4.2 Declarative Annotation and Adjudication Pipeline

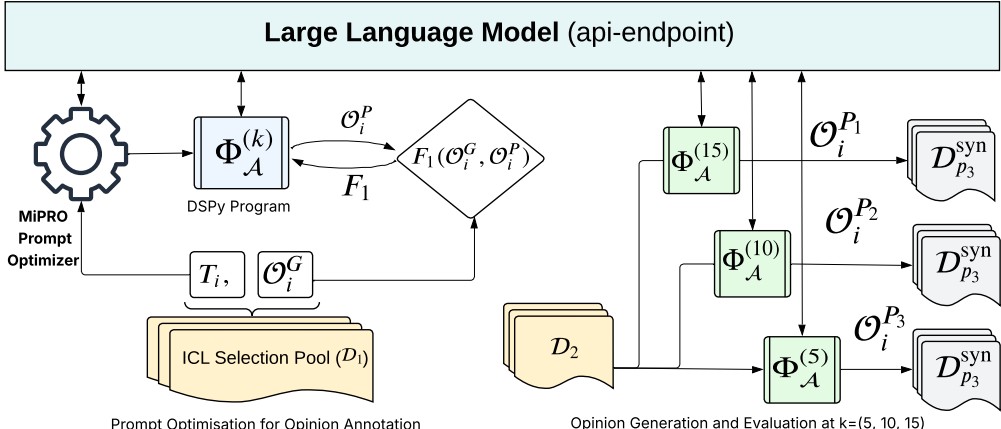

Figure 2: Overview of the LLM-based annotation architecture using DSPy. **Left:** The synthesis stage, where the DSPy program $\Phi_{\mathcal{A}}$ utilizes $k$ human-annotated examples $\mathcal{O}_i^G$ from an ICL selection pool $\mathcal{D}_1$ to construct optimized prompts. **Right:** The annotation stage, where predicted opinions $\mathcal{O}_i^P$ are generated for input texts $T_i$ within the synthetic dataset $\mathcal{D}^{syn}$. The pipeline is evaluated across varying exemplar-counts $k \in \{5, 10, 15\}$ on a distinct development split $\mathcal{D}_2$.

We structure the generation and adjudication steps into a multi-stage declarative pipeline powered by a DSPy program. Both the annotation and adjudication stages rely on the MIPROv2 prompt optimiser to systematically refine the instructions and in-context examples (Opsahl-Ong et al., 2024). The pipeline is trained and evaluated across two data splits: $D_1$ (the first half of the development set) and $D_2$ (the second half of the development set).

**Stage 1: Optimising the Annotation Pipeline on $D_1$** In the first stage, the objective is to optimise the LLM-based opinion generation pipeline, denoted as $\Phi_{\mathcal{A}}^{(k)}$, to extract opinions from the input text $T_i$. We utilise MIPROv2 on $D_1$ to systematically optimise prompts across three few-shot levels, $k \in \{5, 10, 15\}$. The optimiser explores different combinations of instructions and in-context examples sampled from $D_1$ to maximise the quality of the extracted opinions.

**Stage 2: Redundant Generation and Evaluation on $D_2$** Once the generation programs are optimised, we apply them to the unseen texts in $D_2$ to fulfil two purposes. First, we evaluate the performance of the generated pipelines at the different $k$ values by computing their $F_1$ scores against the ground truth annotations, $F_1(\mathcal{O}_i^G, \mathcal{O}_i^p)$, allowing us to determine the optimal $k$ setting.

Second, we use these optimised programs to perform redundant data generation over $D_2$. For a given input text $T_i$, this process yields multiple sets of predicted candidate annotations (denoted as $\mathcal{O}_i^{P_1}, \mathcal{O}_i^{P_2}, \mathcal{O}_i^{P_3}$). This results in a comprehensive dataset containing the original text, redundant LLM predictions, and gold annotations.

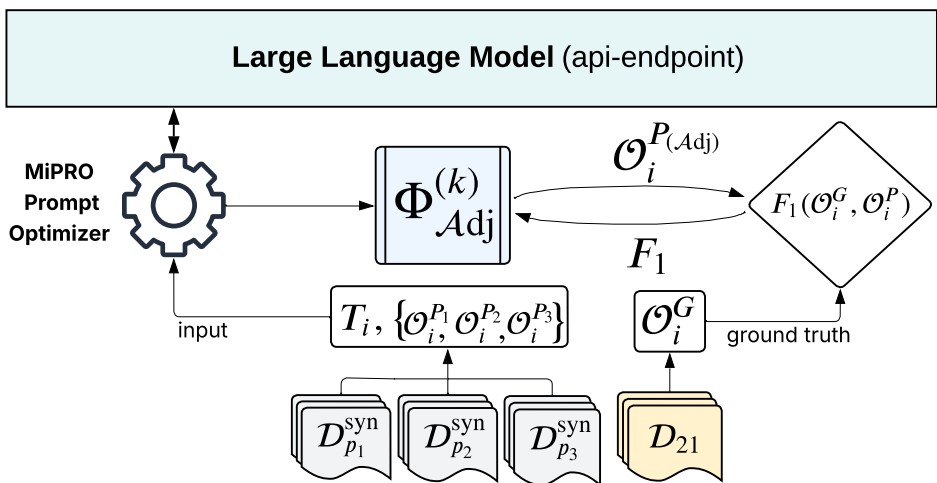

Prompt Optimisation for Opinion Adjudication

Figure 3: LLM-based adjudication DSPy pipeline. Annotations at $k \in \{5, 10, 15\}$ are combined with the original split ($\mathcal{D}_2$) to generate the adjudication DSPy program ($\Phi_{Adj}^{(k)}$)

**Stage 3: Optimising the Adjudication Pipeline** In the final stage, we train the adjudication pipeline ($\Phi_{\mathcal{A}dj}$). This phase exclusively utilises the data formulated in Stage 2 from the $D_2$ split. The inputs to the adjudication model include the original source text $T_i$ and the redundant predicted sets $\{\mathcal{O}_i^{P_1}, \mathcal{O}_i^{P_2}, \mathcal{O}_i^{P_3}\}$.

Using MIPROv2, the adjudicator is optimised to evaluate and resolve conflicts among the redundant opinions to produce a final, synthesised output prediction, $\mathcal{O}_i^{P(\mathcal{A}dj)}$. The optimisation is driven by maximising the $F_1$ score between the final adjudicated output and the ground truth $\mathcal{O}_i^G$ present in $D_2$.

## 5 Experimental Setup

**Model Configuration** In our experiments, we used models of three different sizes: mini ( 4B parameters), small ( 14B parameters), and medium ( 32B parameters). The assignment of LLMs as $\mathcal{A}_1, \mathcal{A}_2, \mathcal{A}_3$ is carried

out based on the model's performance on evaluation samples, and for all cases, the adjudication is performed by $\mathcal{A}_1$, which is the notation assigned to the best-performing LLM-based annotator.

Table 2: LLMs used as annotators

| Annotator | Mini ( 4B) | Small ( 14B) | Medium ( 32B) |
|---|---|---|---|
| $\mathcal{A}_1$ | Qwen3-4B-Thinking | Qwen3-14B | Qwen/Qwen3-30B-A3B-Thinking-2507-FP8 |
| $\mathcal{A}_2$ | MiniCPM3-4B | openai/gpt-oss-20b | unsloth/DeepSeek-R1-Distill-Qwen-32B-bnb-4bit |
| $\mathcal{A}_3$ | Phi-4-mini-reasoning | DeepSeek-R1-Distill-Qwen-14B | openai/gpt-oss-20b |

We evaluate the pipeline across three model scales: **Mini**, **Small**, and **Medium**. Within each scale, we consider three distinct LLMs (see Table 2). For each model, we first identify its optimal configuration by evaluating performance at $k \in \{5, 10, 15\}$ on the $D_2$ split. The models are then ranked as $\mathcal{A}_1, \mathcal{A}_2$, and $\mathcal{A}_3$ based on these $F_1$ scores. Consequently, $\mathcal{A}_n$ represents a specific model-shot combination that maximises that model's extraction capability.

The LLM engine assigned to $\mathcal{A}_1$ is also selected to power the adjudication stage because of its demonstrated efficacy on the annotation task. The adjudication process is analogous to appointing a high-performing human annotator as adjudicator(Hovy & Lavid, 2010). To achieve this, we train a separate DSPy program and then optimise via MIPROv2, obtaining $\mathcal{A}$dj. This two-stage approach ensures that, while the redundant annotations are generated by a diverse ensemble, the final synthesis is performed by the most capable model in the tier, using a prompt specifically optimised to resolve inter-model disagreements.

**Hardware Usage**   Each experiment was run on a single Nvidia A40 GPU, with vLLM (Kwon et al., 2023) used to serve each model through the various steps of our pipeline, as required by DSPy for OpenAI-compliant API access. We keep the LLM hyperparameters constant across all models. The temperature is set to 0.0 to ensure the most deterministic generation. The model's context window determines the input sequence length, and the generated sequence's output length is set to 16384.

**Data Usage**   As detailed in Section X (Method), prompt optimisation and evaluation strictly utilise the development set (partitioned into $\mathcal{D}_1$ and $\mathcal{D}_2$), ensuring that the LLMs are never fine-tuned and that the training data remains completely unseen.

**Evaluation**   We evaluate the pipeline's performance on the ASTE and ACOS tasks using standard exact-match $F_1$ Scores. Results are reported by model scale (Mini, Small, and Medium) to analyse the impact of parameter count on the efficacy of extraction and adjudication.

Exact-match metrics are notoriously stringent for combinatorial tasks that require perfect alignment of spans, categories, and sentiment polarities (Barnes et al., 2021). In our evaluation, we provide a more nuanced error analysis through granular sub-task metrics. Specifically, we report **element-wise** and **pairwise** $F_1$-scores, alongside **Inter-Annotator Agreement (IAA)** for span co-extractions. These additional metrics allow us to determine whether pipeline failures occur at the boundary-boundary-detection level or during the complex relational linking required for triplets and quadruples.

**Baseline Method**   We adopt three foundational SFT paradigms as baselines: sequence tagging, grid-based span labelling, and encoder-decoder generation. These paradigms serve as the primary architectural foundations for supervised ASTE and ACOS methods. Within the proposed DSPy-based framework, foundational SFT models provide the most relevant point of comparison, enabling contextualization of emergent large language model (LLM) capabilities relative to task-specific supervision.

## 6   Results

The trends observed in the LLM-based annotations, compared with the human annotations, are discussed individually for both annotation tasks. The evaluation presented in this section reports the triplet and

quadruplet F1-scores for ASTE and ACOS, respectively. These scores are computed using human annotations as a reference and reflect their alignment with LLM-based annotations.

## 6.1 ASTE

Table 3: LLM-based annotator performances on ASTE (P / R / F1) using Inference-time adaptation methods. The baselines are SFT approaches using: sequence tagging method (BERT-JET by Xu et al. (2020)), grid tagging method (BERT-GTS by Wu et al. (2020) and sequence-to-sequence generative method by Yan et al. (2021).

| | Model | Lap14 P / R / F1 | Res14 P / R / F1 | Res15 P / R / F1 | Res16 P / R / F1 |
|---|---|---|---|---|---|
| **SFT** | JET | 55.39 / 47.33 / 51.04 | 62.27 / 64.98 / 63.86 | 64.45 / 51.96 / 57.53 | 70.42 /58.37 / 63.83 |
| | GTS | 42.34 / 43.39 / 43.15 | 62.27 / 64.98 / 63.86 | 50.96 / 60.60 / 55.11 | 62.43 / 70.81 / 66.36 |
| | BART$_{ABSA}$ | 61.41 / 56.19 / **58.69** | 65.52 / 64.99 / 65.25 | 59.14 / 59.38 / 59.26 | 66.60 / 68.68 / 67.62 |
| **Inference-Time Adaptation** — Mini | $\mathcal{A}_1$ | 42.34 / 43.39 / 43.15 | 62.27 / 64.98 / 63.86 | 50.96 / 60.60 / 55.11 | 62.43 / 70.81 / 66.36 |
| | $\mathcal{A}_2$ | 34.05 / 38.08 / 35.95 | 54.71 / 60.16 / 57.31 | 41.70 / 55.46 / 47.61 | 52.42 / 67.51 / 59.01 |
| | $\mathcal{A}_3$ | 21.63 / 25.92 / 23.59 | 36.18 / 41.22 / 38.53 | 30.12 / 40.87 / 34.68 | 40.65 / 51.27 / 45.34 |
| | $\mathcal{A}_{dj}$ | 45.95 / 45.43 / 45.69 | 67.44 / 67.85 / 67.64 | 54.31 / 62.65 / 58.18 | 64.28 / 71.92 / **67.89** |
| Small | $\mathcal{A}_1$ | 44.05 / 45.85 / 44.93 | 62.39 / 64.49 / 63.42 | 54.92 / 66.80 / 60.28 | 60.33 / 70.43 / 64.99 |
| | $\mathcal{A}_2$ | 40.07 / 42.14 / 41.08 | 57.00 / 63.48 / 60.06 | 47.49 / 62.47 / 53.96 | 53.20 / 66.15 / 58.98 |
| | $\mathcal{A}_3$ | 31.79 / 34.20 / 32.95 | 55.26 / 59.76 / 57.42 | 45.88 / 56.29 / 50.56 | 57.66 / 68.09 / 62.44 |
| | $\mathcal{A}_{dj}$ | 38.89 / 41.77 / 40.24 | 59.13 / 64.38 / 61.65 | 50.66 / 63.56 / 56.38 | 57.01 / 69.64 / 62.69 |
| Med | $\mathcal{A}_1$ | 43.93 / 47.50 / 45.64 | 66.38 / 71.12 / 68.67 | 56.77 / 68.67 / 61.98 | 59.87 / 71.98 / 65.37 |
| | $\mathcal{A}_2$ | 42.31 / 44.73 / 43.49 | 60.38 / 64.69 / 62.46 | 55.03 / 66.60 / 60.26 | 58.25 / 68.68 / 63.04 |
| | $\mathcal{A}_3$ | 40.07 / 42.14 / 41.08 | 57.00 / 63.48 / 50.06 | 47.49 / 62.47 / 53.96 | 53.20 / 66.15 / 58.98 |
| | $\mathcal{A}_{dj}$ | 45.14 / 47.50 / 46.81 | 68.81 / 70.42 / **69.27** | 57.66 / 68.24 / **62.51** | 62.33 / 72.76 / 67.14 |

**Overview** The effectiveness of LLM-based annotators is defined by a critical asymmetry across domains. In the laptop domain (Lap14), domain-specific optimisation appears paramount, as supervised learning remains superior. Here, BART-ABSA achieves 58.69% F1, outperforming the best LLM-based approach (Med $\mathcal{A}dj$, 46.81%) by over 10%. Conversely, this trend reverses in the restaurant domains (Res14, Res15, Res16), where inference-time adaptation methods leverage emergent reasoning to reach up to 69.27% F1. These models consistently outperform all SFT baselines, suggesting that LLMs offer better generalisation through emergent knowledge despite their previously noted relational struggles.

**LLM-As-ASTE Annotators** What the F1 score essentially measures here is the alignment of extracted ASTE triplets with those annotated by humans. The LLM-based annotator methods are categorised into three size-based tiers: There is a measurable increase in $F1$, indicating increased alignment with human-annotated sentiment triplets as model capacity increases from **Mini** to **Med**. We see an increased alignment across all datasets, from Mini to Med, when the results undergo adjudication. The F1 score further increases by +2.54%, +1.95%, +3.07% and 1.53% for the Lap14, Res14, Res15 and Res16 datasets, respectively, for Mini ($\leq$ 4-B) models between the best individual performing annotator model and the adjudicated output, i.e., $\mathcal{A}dj - \mathcal{A}_1$ . For Med ($\leq$ 32B) models we see similar gains of +1.47%, +0.60%, +0.53% and +2.27% across Lap14, Res14, Res15 and Res16 datasets, respectively.

**Trend Anomaly** Adjudication surprisingly reduces alignment for Small models (using Qwen3-14B). We believe this performance deficit suggests that specialised reasoning distillation can outweigh raw parameter scaling for reasoning-intensive tasks (Guo et al., 2025). All other successful adjudicators, at both the mini and med scales, utilise long-chain reasoning states (i.e., "thinking" models). As the experimental results indicate, this capacity for internal reasoning is beneficial for the complex task of annotation adjudication, whereas models lacking this specific optimisation struggle to resolve conflicting relational structures.

Table 4: LLM-based annotator performances on ACOS (P / R / F1) using inference-time adaptation method. The baselines include a rule-based method (Qiu et al., 2011), sequence-labelling methods JET-ACOS and Extract-classify(Cai et al., 2021) and sequence-to-sequence generation method (BART-CRN by (Xiong et al., 2023)).

| | Model | Laptop | Restaurant |
|---|---|---|---|
| | | P / R / F1 | P / R / F1 |
| **SFT** | JET-ACOS | 44.52 / 16.25 / 23.81 | 59.81 / 28.94 / 39.01 |
| | Extract-classify | 45.56 / 29.48 / 35.80 | 38.54 / 52.96 / 44.61 |
| | BART-CRN | 48.16 / 31.83 / **38.32** | 50.84 / 47.10 / **48.90** |
| **Inference-Time Adaptation** | Double-Propagation | 13.04 / 00.57 / 08.00 | 34.67 / 15.08 / 21.04 |
| | Mini $\mathcal{A}_1$ | 13.18 / 12.97 / 13.07 | 32.11 / 28.71 / 30.31 |
| | Mini $\mathcal{A}_2$ | 11.65 / 12.63 / 12.12 | 21.27 / 21.51 / 21.39 |
| | Mini $\mathcal{A}_3$ | 01.36 / 01.90 / 01.58 | 04.62 / 05.89 / 05.18 |
| | Mini $\mathcal{A}_{\mathrm{dj}}$ | 16.71 / 16.17 / 16.43 | 32.90 / 30.45 / 31.63 |
| | Small $\mathcal{A}_1$ | 09.78 / 09.60 / 09.69 | 22.67 / 20.19 / 21.36 |
| | Small $\mathcal{A}_2$ | 07.62 / 07.69 / 07.66 | 24.27 / 21.72 / 22.92 |
| | Small $\mathcal{A}_3$ | 06.92 / 06.49 / 06.70 | 16.88 / 14.74 / 15.73 |
| | Small $\mathcal{A}_{\mathrm{dj}}$ | 10.94 / 11.07 / 11.01 | 30.86 / 29.14 / 29.98 |
| | Med $\mathcal{A}_1$ | 15.77 / 15.74 / 15.75 | 29.37 / 27.51 / 28.41 |
| | Med $\mathcal{A}_2$ | 09.63 / 09.94 / 09.78 | 17.48 / 17.46 / 17.47 |
| | Med $\mathcal{A}_3$ | 07.62 / 07.69 / 07.66 | 24.27 / 21.72 / 22.92 |
| | Med $\mathcal{A}_{\mathrm{dj}}$ | 18.31 / 19.72 / 18.99 | 41.12 / 39.95 / *40.53* |

## 6.2 ACOS

**Overview** For the ACOS quadruple extraction task, the results in Table 4 demonstrate a performance gap that highlights the distinction between learnt pattern recognition and inference-time methods. The SFT baselines, particularly **BART-CRN** (38.32% on Laptop and 48.90% on Restaurant), maintain a noticeable lead.

Unlike SFT models, inference-time adaptation methods (LLM-based Annotators) cannot rely on learned statistical likelihoods to navigate these two primary challenges:

- **Presence of Implicit Opinions and Aspects**: The transition from triplets to quadruples introduces a decision-making layer that goes beyond span extraction; the model must determine whether an aspect or opinion is explicit or implicit, then map it to an exact category.

- **Search Space Complexity**: In the Laptop dataset, the taxonomy consists of 112 unique categories. For LLM-based annotation methods, this high-entropy choice space yields the best F1 score of only 18.99% while the Restaurant dataset, featuring only 12 categories, presents a much simpler classification task. Here, proposed LLM-based annotation methods are more effective, with the Med tier achieving a competitive $F1$ score of 40.53%.

**LLM-As-ACOS Annotators** Despite increased complexity in identifying implicit elements, we still observe measurable $F1$ gains through adjudication (where results from multiple models are consolidated, denoted as $\mathcal{A}_{\mathrm{dj}}$ − Best Individual) in both the Mini and Med model tiers. Specifically, Mini models show $F1$ score increases of +3.36% and +1.32% for the Laptop and Restaurant datasets, while the Med tier achieves even higher gains of +3.24% and +12.12%.

We observe a similar trend to that observed in the ASTE results in the Small tier. Here, adjudication improves performance but still yields the lowest adjudicated $F1$ scores: only 11.01% and 29.98%. As discussed earlier, reliance on Qwen3-14B, which lacks specialised reasoning distillation, is a bottleneck. Larger parameters usually help memorise taxonomies in SFT. But with LLM-based methods that rely on inference-time adaptation, the absence of long-chain reasoning states leads to lower overall $F1$ scores.

## 7   Analysis

The element-wise and pairwise decomposition of triplet and quadruple evaluations, detailed in Tables 5 and 6, reveals a consistent pattern across both tasks. Alignment with human annotations is strong at the individual span level but degrades systematically as relational complexity increases. Decomposing the joint evaluation of triplets (ASTE) and quadruples (ACOS) into their constituent labels allows us to identify where this degradation occurs and quantify the relative difficulty of interdependent subtasks.

Table 5: Element-wise assessment of ASTE's alignment with human annotations (F1 score).

| Data | $\mathcal{A}$ | Mini | | | | | | Small | | | | | | Med | | | | | |
|---|---|---|---|---|---|---|---|---|---|---|---|---|---|---|---|---|---|---|---|
| | | s | at | op | s&at | s&op | at&op | s | at | op | s&at | s&op | at&op | s | at | op | s&at | s&op | at&op |
| lap14 | $\mathcal{A}_1$ | 83.55 | 66.74 | 64.02 | 58.22 | 56.53 | 48.67 | 85.80 | 66.18 | 65.93 | 59.56 | 60.36 | 48.91 | 84.81 | 66.12 | 72.73 | 58.02 | 64.64 | 51.33 |
| | $\mathcal{A}_2$ | 80.81 | 63.21 | 50.98 | 53.82 | 45.61 | 39.82 | 83.29 | 64.30 | 61.88 | 55.96 | 53.97 | 47.03 | 85.80 | 64.46 | 68.75 | 56.49 | 60.52 | 49.96 |
| | $\mathcal{A}_3$ | 79.77 | 51.34 | 37.39 | 43.30 | 32.01 | 26.99 | 86.25 | 58.32 | 49.31 | 52.30 | 44.12 | 36.87 | 83.29 | 64.30 | 61.88 | 55.96 | 53.97 | 47.03 |
| | Adj | 83.46 | 67.38 | 69.24 | 57.62 | 61.55 | 52.62 | 84.21 | 63.81 | 60.43 | 56.07 | 53.82 | 45.63 | 84.44 | 67.36 | 72.91 | 58.03 | 64.37 | 53.37 |
| res14 | $\mathcal{A}_1$ | 90.86 | 79.93 | 75.50 | 75.64 | 71.77 | 67.13 | 89.94 | 78.35 | 77.00 | 73.95 | 73.06 | 66.43 | 90.44 | 81.35 | 79.38 | 77.53 | 75.73 | 71.10 |
| | $\mathcal{A}_2$ | 85.61 | 77.74 | 68.55 | 70.72 | 62.91 | 61.69 | 88.21 | 77.08 | 72.02 | 72.48 | 67.76 | 62.42 | 89.53 | 78.65 | 75.79 | 73.73 | 71.70 | 65.86 |
| | $\mathcal{A}_3$ | 86.49 | 63.91 | 53.05 | 59.00 | 49.63 | 40.38 | 90.50 | 77.81 | 68.51 | 73.69 | 65.24 | 59.86 | 88.19 | 77.35 | 72.29 | 72.75 | 67.97 | 62.83 |
| | Adj | 90.63 | 80.47 | 79.97 | 75.97 | 76.05 | 70.59 | 90.24 | 78.56 | 73.58 | 74.11 | 69.96 | 64.78 | 90.20 | 81.64 | 79.82 | 77.28 | 75.98 | 72.24 |
| res15 | $\mathcal{A}_1$ | 88.49 | 76.80 | 72.87 | 69.92 | 64.91 | 61.19 | 90.53 | 82.39 | 73.86 | 75.62 | 67.52 | 66.11 | 89.50 | 77.44 | 78.25 | 71.00 | 71.30 | 67.79 |
| | $\mathcal{A}_2$ | 83.26 | 75.46 | 63.20 | 66.19 | 54.44 | 54.37 | 88.71 | 78.43 | 68.71 | 71.80 | 61.82 | 59.70 | 89.47 | 78.92 | 76.62 | 72.42 | 69.38 | 66.04 |
| | $\mathcal{A}_3$ | 85.96 | 60.81 | 51.07 | 53.82 | 46.01 | 38.76 | 91.01 | 79.43 | 65.75 | 73.85 | 59.06 | 55.76 | 88.74 | 78.52 | 68.83 | 71.92 | 61.96 | 59.84 |
| | Adj | 88.48 | 79.02 | 76.14 | 71.10 | 68.37 | 65.13 | 89.85 | 80.56 | 70.76 | 73.62 | 63.94 | 62.63 | 89.69 | 79.82 | 78.27 | 73.73 | 71.63 | 67.99 |
| res16 | $\mathcal{A}_1$ | 88.75 | 81.83 | 78.93 | 76.14 | 73.16 | 71.23 | 89.79 | 79.35 | 79.61 | 73.74 | 74.29 | 69.48 | 88.98 | 78.83 | 80.55 | 73.16 | 74.68 | 70.14 |
| | $\mathcal{A}_2$ | 88.83 | 75.00 | 72.00 | 69.78 | 67.05 | 62.69 | 87.14 | 76.36 | 75.34 | 70.04 | 69.56 | 63.49 | 90.91 | 76.98 | 78.16 | 72.18 | 73.35 | 66.79 |
| | $\mathcal{A}_3$ | 87.91 | 65.34 | 57.80 | 61.07 | 53.83 | 47.58 | 89.97 | 79.01 | 74.42 | 74.11 | 70.54 | 65.83 | 87.14 | 76.36 | 75.34 | 70.04 | 69.56 | 63.49 |
| | Adj | 89.55 | 81.96 | 81.33 | 76.29 | 75.70 | 72.49 | 89.04 | 78.09 | 77.21 | 72.44 | 71.77 | 67.08 | 89.73 | 80.04 | 82.44 | 74.62 | 76.45 | 72.17 |

Table 6: Element-wise assessment of ACOS alignment with human annotations (F1 score).

| Dataset | Scale & Annot | | s | at | op | E | A | ac | $\frac{s}{+at}$ | $\frac{s}{+op}$ | $\frac{at}{+op}$ | $\frac{s}{+ac}$ | $\frac{at}{+ac}$ | $\frac{op}{+ac}$ | $\frac{at+}{s+op}$ |
|---|---|---|---|---|---|---|---|---|---|---|---|---|---|---|---|
| laptop | small | $\mathcal{A}_1$ | 89.48 | 68.97 | 57.05 | 53.05 | 54.05 | 30.75 | 63.47 | 54.96 | 42.50 | 28.26 | 21.68 | 18.06 | 40.65 |
| | | $\mathcal{A}_2$ | 88.52 | 62.30 | 45.09 | 66.95 | 54.12 | 34.99 | 56.61 | 42.59 | 30.39 | 31.43 | 23.54 | 18.19 | 28.55 |
| | | $\mathcal{A}_3$ | 73.39 | 37.60 | 17.16 | 46.63 | 39.58 | 19.53 | 32.25 | 15.98 | 10.42 | 16.43 | 7.24 | 3.40 | 9.91 |
| | | Adj | 89.70 | 68.45 | 55.21 | 67.81 | 54.83 | 39.87 | 63.12 | 52.30 | 40.37 | 36.29 | 27.76 | 22.75 | 38.50 |
| Restaurant | | $\mathcal{A}_1$ | 90.34 | 68.69 | 65.58 | 62.51 | 76.59 | 55.87 | 63.70 | 61.79 | 51.20 | 51.85 | 40.53 | 39.00 | 48.75 |
| | | $\mathcal{A}_2$ | 87.77 | 61.60 | 55.47 | 57.80 | 65.34 | 49.39 | 58.54 | 52.76 | 41.05 | 45.86 | 31.35 | 29.70 | 39.51 |
| | | $\mathcal{A}_3$ | 76.86 | 38.22 | 23.84 | 34.68 | 49.03 | 26.51 | 34.30 | 21.89 | 15.26 | 23.54 | 13.85 | 8.46 | 14.27 |
| | | Adj | 90.48 | 69.26 | 58.19 | 85.88 | 75.92 | 66.25 | 64.77 | 54.08 | 45.64 | 61.83 | 50.14 | 40.86 | 43.00 |
| laptop | med | $\mathcal{A}_1$ | 90.88 | 59.26 | 36.86 | 65.58 | 59.28 | 40.56 | 55.24 | 35.22 | 22.85 | 37.48 | 23.66 | 16.83 | 21.88 |
| | | $\mathcal{A}_2$ | 89.30 | 57.66 | 44.10 | 49.61 | 49.31 | 21.97 | 53.06 | 41.23 | 27.21 | 19.92 | 15.54 | 10.62 | 25.65 |
| | | $\mathcal{A}_3$ | 90.62 | 63.35 | 38.24 | 48.28 | 50.50 | 21.86 | 58.98 | 35.99 | 25.16 | 20.13 | 13.67 | 10.19 | 23.89 |
| | | Adj | 90.85 | 60.04 | 34.75 | 76.28 | 60.71 | 46.74 | 55.77 | 32.77 | 22.59 | 43.32 | 27.08 | 18.10 | 21.37 |
| restaurant | | $\mathcal{A}_1$ | 90.45 | 57.85 | 46.60 | 67.11 | 78.10 | 58.95 | 54.22 | 44.18 | 34.11 | 54.98 | 37.52 | 27.80 | 32.59 |
| | | $\mathcal{A}_2$ | 87.93 | 60.00 | 49.27 | 54.07 | 68.22 | 49.11 | 56.03 | 46.46 | 37.32 | 44.55 | 36.23 | 27.91 | 35.69 |
| | | $\mathcal{A}_3$ | 90.26 | 64.57 | 50.73 | 51.98 | 67.26 | 43.78 | 60.42 | 46.77 | 36.58 | 39.51 | 28.45 | 22.93 | 34.34 |
| | | Adj | 90.08 | 64.18 | 47.57 | 79.63 | 84.32 | 71.48 | 59.87 | 44.72 | 37.71 | 66.00 | 49.19 | 37.46 | 35.78 |
| laptop | large | $\mathcal{A}_1$ | 89.86 | 68.20 | 55.29 | 62.96 | 53.05 | 36.30 | 62.67 | 52.17 | 39.57 | 33.35 | 25.95 | 22.61 | 37.46 |
| | | $\mathcal{A}_2$ | 90.37 | 61.84 | 43.96 | 56.42 | 56.75 | 33.15 | 57.22 | 42.07 | 29.04 | 30.59 | 20.72 | 15.40 | 27.66 |
| | | $\mathcal{A}_3$ | 89.30 | 57.66 | 44.10 | 49.61 | 49.31 | 21.97 | 53.06 | 41.23 | 27.21 | 19.92 | 15.54 | 10.62 | 25.65 |
| | | Adj | 90.40 | 73.53 | 59.59 | 70.51 | 54.27 | 41.23 | 67.71 | 56.60 | 45.80 | 37.90 | 30.68 | 24.84 | 43.53 |
| restaurant | | $\mathcal{A}_1$ | 89.94 | 69.39 | 64.13 | 65.41 | 72.28 | 54.91 | 63.90 | 60.16 | 50.09 | 49.01 | 39.26 | 37.66 | 47.46 |
| | | $\mathcal{A}_2$ | 91.50 | 63.30 | 54.71 | 52.97 | 63.72 | 46.25 | 58.98 | 52.52 | 39.54 | 42.48 | 28.08 | 25.09 | 38.32 |
| | | $\mathcal{A}_3$ | 87.93 | 60.00 | 49.27 | 54.07 | 68.22 | 49.11 | 56.03 | 46.46 | 37.32 | 44.55 | 36.23 | 27.91 | 35.69 |
| | | Adj | 91.01 | 72.74 | 64.60 | 82.96 | 82.43 | 71.75 | 68.13 | 61.17 | 52.86 | 66.40 | 55.78 | 48.85 | 50.06 |

**Performance Hierarchy and Label Alignment** : Across both tasks, sentiment polarity (s) consistently exhibit the highest alignment with human annotations, frequently exceeding 90% F1. Aspect term (*at*) and opinion term (*op*) spans show lower, yet comparable, alignment scores. However, performance declines predictably as models are required to identify interdependent structures. In the ASTE task (Table 5), while assigning the correct polarity to an aspect (*s&at*) is more challenging than individual labelling, the most degrading bottleneck appears to be *at&op* identification, i.e. the process of correctly linking an opinion span to its specific target.

**Domain Disparity and Complexity in ACOS** Similar trends persist in the ACOS task (Table 6), though the increased dimensionality introduces further complexity. Models exhibit a pronounced domain disparity, particularly in the laptop domain, where the lack of dataset-specific tuning appears to hinder performance. This difficulty is largely due to the prediction of aspect categories (ac), which reinforces the previously discussed challenge of increased search-space complexity for *ac*, with 112 categories in the Laptop domain and 12 in the restaurant domain. The lower scores for the aspect term and opinion span than for the ASTE counterparts can be explained by the presence of implicit sentiments and aspects, which must be left unlabelled (or set to None) when labelling quadruples.

**Adjudication Effects** The adjudication mechanism consistently mitigates errors made by individual LLM-based annotators. The effect is most notable in the high-dimensional ACOS task, where adjudication helps resolve structural ambiguities. Overall, these findings indicate that while LLMs provide a robust foundation for automated labelling, aligning interdependent relational labels remains a primary area for further refinement.

## 7.1 Inter-Annotator Agreement of LLM-Based Annotators

Table 7: Top: Krippendorff-$\alpha$ for IAA for ASTE span annotations (*at*: Aspect, *op*: Opinion, *at&s*: Aspect Term with sentiment, *op&s*: Opinion span with sentiment). Bottom: Performance scores for ACOS tasks (*ac*: Aspect Category).

| Model | lap14 | | | | res14 | | | | res15 | | | | res16 | | | |
|-------|------|------|------|------|------|------|------|------|------|------|------|------|------|------|------|------|
| | *at* | *op* | *at&s* | *op&s* | *at* | *op* | *at&s* | *op&s* | *at* | *op* | *at&s* | *op&s* | *at* | *op* | *at&s* | *op&s* |
| Mini | 0.6125 | 0.49 | 0.59 | 0.46 | 0.70 | 0.61 | 0.68 | 0.59 | 0.70 | 0.57 | 0.67 | 0.54 | 0.72 | 0.63 | 0.69 | 0.60 |
| Small | 0.73 | 0.66 | 0.72 | 0.66 | 0.79 | 0.74 | 0.78 | 0.74 | 0.81 | 0.72 | 0.80 | 0.71 | 0.79 | 0.77 | 0.78 | 0.75 |
| Med | 0.75 | 0.69 | 0.75 | 0.69 | 0.82 | 0.77 | 0.80 | 0.77 | 0.84 | 0.75 | 0.82 | 0.74 | 0.82 | 0.79 | 0.81 | 0.78 |

| Model | Laptop | | | | | Restaurant | | | | |
|-------|------|------|------|------|------|------|------|------|------|------|
| | *at* | *op* | *ac* | *at&s* | *op&s* | *at* | *op* | *ac* | *at&s* | *op&s* |
| Mini | 0.60 | 0.37 | 0.22 | 0.58 | 36.75 | 0.61 | 0.41 | 0.41 | 0.59 | 0.40 |
| Small | 0.70 | 0.65 | 0.31 | 0.69 | 0.65 | 0.73 | 0.64 | 0.49 | 0.71 | 0.64 |
| Med | 0.70 | 0.64 | 0.33 | 0.69 | 0.64 | 0.71 | 0.65 | 0.50 | 0.69 | 0.64 |

The joint evaluation reported in table 3 shows promising results, but the outcome of the annotation pipeline for ACOS reported in table 4 forces the conclusion that LLM-based pipelines are not yet capable of performing complex annotation individually. However, upon analysis of individual sentiments and spans (Tables 5 and 6), we observe promising results hinting at potential span-level annotation capabilities that warrant further investigation for reliability. In accordance with standard annotation practices, we use and report the pre-adjudication Inter-Annotator Agreement (IAA) to assess the reliability of LLM-based annotators. While F1-scores were used for the human-annotated dataset used in our experiments, their limitations as an IAA metric are noted in (Klie et al., 2024). Thus, we employ Krippendorff's $\alpha$ scores to assess the IAAs of our LLM-based span annotation.

Upon examining the Krippendorff $\alpha$ values, we observe that the reliability of the labels for ASTE and ACOS increases with model parameter size. For ASTE, we see the reliability of triplet span elements approaching

**highly reliable levels**, whereas ACOS quadruple elements tend to be **moderately reliable** at best. The aforementioned trend highlights the observed increase in annotation reliability with increasing model size.

## 8 Error Analysis

The qualitative evidence suggests that ACOS and ASTE tasks are inherently subjective, complicating the notion of a singular "correct" extraction. While the transition to ASTE simplifies the task by omitting aspect categories, it remains susceptible to divergent readings of narrative context. For example, in the sentence *"The government's plan to suppress the protests was met with public outrage"*, the inclusion of *"protests"* as a negative signal or "plan" as a distinct target highlights a more granular interpretative lens than that adopted by the Gold standard.

Table 8: Qualitative ACOS (quadruple) and ASTE (triplet) annotation examples. ACOS cells display the tuple (*s*: sentiment, *at*: aspect term, *ac*: aspect category, *op*: opinion span). ASTE rows omit the category field, using *t* for target and *o* for opinion. NULL indicates an implicit element. Incorrect entries are highlighted relative to the Gold annotation.

| [**ACOS**] **Text 1:** *It freezes constantly and needs to be shut off to start again.* | | | | |
|---|---|---|---|---|
| **Gold:**
*s*: Negative
*at*: NULL
*ac*: laptop#op_perf
*op*: NULL | **𝒜₁:**
*s*: Negative
*at*: NULL
*ac*: laptop#op_perf
*op*: freezes constantly | **𝒜₂:**
*s*: Negative
*at*: system
*ac*: os#op_perf
*op*: freezes constantly | **𝒜₃:**
*s*: Negative
*at*: freezes
*ac*: cpu#op_perf
*op*: freezes constantly | **Adj:**
*s*: Negative
*at*: NULL
*ac*: laptop#op_perf
*op*: freezes |

| [**ACOS**] **Text 2:** *space was limited , but the food made up for it .* | | | | |
|---|---|---|---|---|
| **Gold:**
*s*: Negative
*at*: space
*ac*: rest#misc
*op*: limited | **𝒜₁:**
*s*: Negative
*at*: space
*ac*: rest#gen
*op*: limited | **𝒜₂:**
*s*: Negative
*at*: space
*ac*: rest#gen
*op*: limited | **𝒜₃:**
*s*: Negative
*at*: space
*ac*: rest#gen
*op*: limited | **Adj:**
*s*: Negative
*at*: space
*ac*: rest#gen
*op*: limited |
| *s*: Positive
*at*: food
*ac*: food#quality
*op*: NULL | *s*: Positive
*at*: food
*ac*: rest#gen
*op*: made up for it | *s*: Positive
*at*: food
*ac*: rest#gen
*op*: made up for it | *s*: Positive
*at*: food
*ac*: food#quality
*op*: made up for it | *s*: Positive
*at*: food
*ac*: food#quality
*op*: made up for it |

| [**ACOS**] **Text 3:** *however , one sunday afternoon my husband and i did go ( although with my loud protests ) and were pleasantly surprised .* | | | | |
|---|---|---|---|---|
| **Gold:**
*s*: Positive
*at*: NULL
*ac*: rest#gen
*op*: pleasantly | **𝒜₁:**
*s*: Positive
*at*: outing
*ac*: loc#gen
*op*: pleasantly | **𝒜₂:**
*s*: Negative
*at*: NULL
*ac*: serv#gen
*op*: loud protests | **𝒜₃:**
*s*: Positive
*at*: trip
*ac*: ship#gen
*op*: pleasantly surprised | **Adj:**
*s*: Positive
*at*: NULL
*ac*: rest#gen
*op*: pleasantly |
| | | *s*: Positive
*at*: NULL
*ac*: serv#gen
*op*: pleasantly surprised | | |

| [**ASTE**] **Text 1:** *tech support would not fix the problem unless I bought your plan for $ 150 plus .* | | | | |
|---|---|---|---|---|
| **Gold:**
*s*: Negative
*t*: tech support
*o*: not fix | **𝒜₁:**
*s*: Negative
*t*: tech support
*o*: would not fix | **𝒜₂:**
*s*: Negative
*t*: tech support
*o*: would not fix | **𝒜₃:**
*s*: Negative
*t*: tech support
*o*: not fix | **Adj:**
*s*: Negative
*t*: tech support
*o*: would not fix |
| | | *s*: Negative
*t*: plan
*o*: plan for $150 plus | | |

| [**ASTE**] **Text 2:** *I liked the atmosphere very much but the food was not worth the price .* | | | | |
|---|---|---|---|---|
| **Gold:**
*s*: Positive
*t*: atmosphere
*o*: liked | **𝒜₁:**
*s*: Positive
*t*: atmosphere
*o*: liked | **𝒜₂:**
*s*: Positive
*t*: atmosphere
*o*: liked | **𝒜₃:**
*s*: Positive
*t*: atmosphere
*o*: liked | **Adj:**
*s*: Positive
*t*: atmosphere
*o*: liked |
| *s*: Negative
*t*: food
*o*: not worth the price | *s*: Negative
*t*: food
*o*: not worth | *s*: Negative
*t*: food
*o*: not worth | *s*: Negative
*t*: food
*o*: not worth the price | *s*: Negative
*t*: food
*o*: not worth |

Also, the constant differences in the boundary of extracted text spans, i.e., the difference between *"not worth"* and *"not worth the price"*, highlight a basic problem with strict evaluation measures. Since Exact Match F1 counts any small difference in text as a wrong answer, it might overstate mistakes by people who understand the meaning but choose different words.

Lastly, the adjudication process helps combine different viewpoints, but it likely produces a shared understanding rather than an absolute truth. This suggests that high error rates in these areas are not just signs of poor performance but may result from a scoring system that struggles to measure answers that might be correct. By analysing errors relative to human-annotated ground truth, we observe that subjective reasoning, one often negotiates meaning rather than simply finding the right answer.

## 9  Limitation

Several limitations warrant consideration, starting with our use of model quantisation to deploy large-scale models on consumer-grade hardware, specifically the 32B variants. While this approach ensures computational efficiency, it may introduce subtle reasoning trade-offs that increase friction in faithfully reproducing relational structures. Secondly, the pipeline's declarative method with inference-time adaptation is theoretically domain-agnostic, our current assessment is limited to SemEval-derived review datasets from the laptop and restaurant domains. Consequently, the pipeline's robustness on more complex or novel unannotated corpora remains a subject for future research. Finally, the process is not strictly annotation-free, as it relies on a small number of in-context learning examples ($k \in \{5, 10, 15\}$). While our method is resource-efficient as it does not require fine-tuning or parametrised optimisation, it still requires a modest foundation of high-quality human labels (least 5-10 examples) to automatically synthesise prompts.

## 10  Conclusion

This work introduces an LLM-based declarative opinion-annotation pipeline and an opinion-adjudication method to resolve inter-annotator disagreements. The declarative annotation pipeline uses ASTE and ACOS schema to enforce extractive constraints on opinions during extraction. Evaluation across multiple model scales, using human-annotated labels as the ground truth, reveals a critical performance bifurcation: although LLMs are effective in detecting individual spans, their structural reliability declines markedly as relational complexity increases. We further assess the reliability of span sentiment co-extraction by calculating Krippendorff's $\alpha$ among the LLM-based annotators. Our findings reveal that LLM-based opinion annotators exhibit strong span-level utility but weaker reliability in annotating the underlying opinion structures that connect those spans. At current levels, the investigated LLMs can provide reliable span-level span and span-sentiment annotations, motivating their use as high-fidelity annotation assistants and data augmentation tools rather than stand-alone opinion annotators.

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
