# OpenReview forum: "Large Language Models as Automatic Annotators and Annotation Adjudicators for Fine-Grained Opinion Analysis"
_TMLR — Under review for TMLR_

### Review · Reviewer_Z1qE · 2026-06-01

**Summary Of Contributions:**

This paper investigates whether LLMs can serve as automatic annotators for fine-grained opinion analysis tasks, specifically Aspect Sentiment Triplet Extraction (ASTE) and Aspect-Category-Opinion-Sentiment (ACOS) quadruple extraction. It proposes a declarative annotation pipeline built on DSPy that uses schema-constrained prompts (optimized via MIPROv2) to extract structured opinion annotations without manual prompt engineering. It also introducse an LLM-based adjudication method that aggregates annotations from multiple LLM annotators (of varying sizes: 4B, 14B, 32B) to produce a final consolidated annotation, analogous to human annotation adjudication. The key empirical finding is that LLMs are reliable at extracting individual spans (aspect terms, opinion terms, sentiment polarity) but degrade substantially when required to reproduce the relational structures that connect those spans into triplets or quadruples.

**Audience:**

Yes

**Audience Explanation:**

NLP/annotation researchers would benefit from the empirical characterization of where LLMs succeed and fail as annotators for structured extraction tasks. The span-vs-relation bifurcation is a practical insight, but the final finding that LLMs cannot fully replace human annotators for complex structured tasks is not particularly surprising.

**Broader Impact Concerns:**

This is no Broader Impact Concerns in the paper and I think the paper domain doesn't apply this statement section necessarily.

**Claims And Evidence:**

Yes

**Claims Explanation:**

It's partially supported.

The central claim about the performance bifurcation (strong span-level, weak relational-level) is well-supported through the element-wise decomposition in Tables 5 and 6, which clearly shows F1 scores dropping as structural complexity increase. The Krippendorff's α analysis in Table 7 further corroborates this by showing that IAA degrades from individual spans to paired elements.

But several claims are less convincingly supported:

1. The adjudication claim is mixed. While adjudication improves over the best individual annotator in most cases (Mini and Med tiers), it
  reduces performance in the Small tier. The inconsistency weakens the general claim that LLM adjudication is a reliable
  mechanism.
2. The comparison to SFT baselines is incomplete. The SFT baselines (BERT-JET, GTS, BART-ABSA, BART-CRN) are from 2020–2023 and use much smaller models. The paper should compare its method against state-of-the-art supervised methods with comparable size.
3. The domain-specific asymmetry claim should be further analyzed. The explanation (112 vs. 12 categories) is plausible but not tested through controlled experiments.

**Requested Changes:**

1. Include stronger and more recent baselines. The SFT baselines are outdated. It could be compared against: (a) fine-tuned LLMs of similar size (e.g., fine-tuned Qwen3-4B/14B on ASTE/ACOS), (b) recent supervised methods for ASTE/ACOS, and (c) larger frontier LLMs (GPT-4o, Claude Sonnet, Llama 70B+) to establish an upper bound for the in-context learning paradigm.

  2. Explain and analyze the Small-tier adjudication failure more rigorously. The claim that Qwen3-14B lacks specialised reasoning
  distillation needs evidence.

  3. Add variance/reporting. Run the pipeline multiple times and report standard deviations on all key metrics. The current results are single-run and may not be reproducible.

  4. Provide a cost-benefit analysis. The paper motivates LLM annotation as addressing the cost/scalability limitations of human annotation, so the comparison of annotation cost vs. human annotation cost or throughput is essential.

---

> ### Author Response · Authors · 2026-06-05
> **Additional results for adjudication claim, and other explanations**
>
> We are grateful for the insights into the work provided by the reviewers. We appreciate that they feel the central claim is well supported by the evidence presented in the draft and try to address the raised concerns.
> 1. **Adjudication Claim**: Adjudication improves F1 across all datasets and tasks except ASTE triplet extraction using a small-scale LLM adjudicator. This issue is strictly isolated to ASTE; adjudication succeeds for all mini/medium scales and across all ACOS configurations.
>    We attributed this drop to the absence of long-chain reasoning in Qwen3-14B, citing Guo et al., 2025 (https://arxiv.org/abs/2501.12948). To empirically validate this claim (as suggested by the reviewer), we ran an ablation study using Qwen-4B-Thinking-2070, a substantially smaller model with explicit long-chain reasoning. Despite its smaller size, this model outperforms Qwen3-14B as an adjudicator (row: Small (\mathcal{A}dj)), directly showing that reasoning depth, not model size, limits adjudication quality.
>
> Only F1 Scores (P,R also available) from additional experiment to support claims at Section 6.1 (paragraph- **Trend Anomaly**)
>
> |                                    | lap14 | res14 | res15 | res16 |
> | ---------------------------------- | ----- | ----- | ----- | ----- |
> | Qwen-14B ($\mathcal{A}dj$)         | 40.24 | 61.65 | 56.38 | 62.69 |
> | Qwen-4B-Thinking ($\mathcal{A}dj$) | 43.28 | 66.46 | 59.94 | 66.48 |
>
> 2. **112 vs 12 categories**: Table 4 shows a clear performance asymmetry between the laptop and restaurant domains for quadruple extraction. Since a quadruple is only counted as correct when all four elements, aspect term (at), opinion span (op), aspect category (ac), and sentiment polarity (s) are predicted correctly, errors in any single element invalidate the entire prediction. This makes category prediction a critical bottleneck, and the two domains differ substantially here: laptop has 112 unique aspect categories, whereas restaurant has only 12.
>
>    We argue this asymmetry is not merely empirical but theoretically principled for two reasons.
>    - First, the combinatorial difficulty of predicting the correct category is about 9 times higher in the laptop domain.
>    - Second, under a uniform distribution, the per-prediction entropy is $log_2(112) \approx 6.81$ bits for the laptop versus $log_2(12) \approx 3.58$ bits for the restaurant, nearly double the uncertainty in the category slot alone.
>
>    This theoretical expectation is directly corroborated by our results in **Table 6**: whenever a joint score includes ac as a component, the performance drop is consistently larger in the laptop domain than in the restaurant, across all model scales. Qualitative evidence of where and how these errors occur is provided in the error analysis (**Table 8**).
>
> 3. **Reproducibility**: We thank the reviewer for noting that our primary claims are well substantiated. We highlight that this support is consistent across the full evidence base rather than a single setting: the main effects hold on both tasks **(ASTE, Table 3; ACOS, Table 4)** and persist under element-wise/combinatoric evaluation (**Tables 5–6**), spanning three model scales (nine pipelines) and six datasets. The narrower claim of span-level annotation capability is supported specifically by inter-annotator agreement in **Table 7**, and **Table 8** studies the failure patterns mentioned in the results. On reproducibility: because our method is an inference-time adaptation rather than fine-tuning, and we set temperature to 0 (Experimental Setup), individual runs are near-deterministic, so standard deviation across repeated runs would likely be near zero and uninformative. We therefore argue that consistency across model scales, tasks, and datasets is a stronger form of reproducibility than variance across repeated runs of a single pipeline.
> 4. **Cost and Model Choices**: We utilise open-weight models (4B–32B) on commodity hardware to achieve a training-free, zero-API-cost inference-time adaptation paradigm. While task-specific fine-tuning or proprietary models (gpt4o,claud) are performant, they contradict our core objective of cost-free, local deployment. Regarding economic benefit, complex tasks like ACOS quadruples typically require expensive, time-consuming human annotation and adjudication rounds. Our pipeline replaces this entirely, bounding marginal costs strictly to local compute time with zero recurring expenses.
>
> 5. **Comparison and Baselines**: We agree to include recent baselines in the revision. Our initial selection (Sequence Tagging, Seq2Seq, Grid-Based) aimed to contextualize our training-free paradigm against dominant foundational SFT approaches, rather than compete directly with fine-tuned SOTA. No equivalent inference-time paradigm exists for ASTE, and ACOS features only weak, rule-based alternatives. Incorporating the latest baselines for direct comparison is straightforward and will be reflected in the updated manuscript.

---

### Review · Reviewer_ZCYT · 2026-06-12

**Summary Of Contributions:**

This paper investigates whether large language models can serve as automatic annotators and annotation adjudicators for fine-grained opinion analysis tasks. The authors focus on two structured sentiment extraction settings, Aspect Sentiment Triplet Extraction (ASTE) and Aspect-Category-Opinion-Sentiment (ACOS). They introduce a declarative annotation pipeline based on DSPy and MIPROv2 that automatically optimizes prompts for opinion extraction without manual prompt engineering. In addition, the paper proposes an LLM-based adjudication mechanism that aggregates annotations produced by multiple LLM annotators to resolve disagreements and generate a final annotation.

The paper evaluates multiple model scales ranging from 4B to 32B parameters and studies both annotation quality and inter-annotator agreement. A key finding is that LLMs perform relatively well at extracting individual spans and sentiment labels but struggle when required to recover higher-order relational structures such as opinion-aspect associations and ACOS quadruples. The authors further demonstrate that adjudication consistently improves annotation quality, particularly for more complex extraction tasks.

Strengths include a clearly defined problem, a reproducible DSPy-based annotation framework, extensive empirical evaluation across multiple datasets, and a thoughtful analysis of where LLM annotation succeeds and fails. The paper also provides useful decomposition analyses and inter-annotator agreement measurements that offer insight beyond aggregate F1 scores.

Weaknesses include limited methodological novelty, as the work primarily combines existing prompting, optimization, and adjudication techniques rather than introducing a fundamentally new learning method. The adjudication framework itself is largely prompt-optimized LLM aggregation, and the empirical study is restricted to SemEval-style review datasets. In addition, the experimental design uses development splits both for prompt optimization and adjudication training, raising concerns regarding possible overfitting to benchmark-specific characteristics.

**Additional Comments:**

I found the paper well organized and substantially stronger than many recent empirical studies on LLM-assisted annotation. The decomposition analysis and inter-annotator agreement evaluation are particularly useful because they help identify where current LLMs fail rather than merely reporting aggregate performance. The central conclusion—that LLMs are effective span annotators but weaker relational annotators—is both plausible and well supported by the reported evidence.

My primary reservation concerns novelty. Much of the technical framework is built from existing components (DSPy, MIPROv2, multiple LLM annotators, and LLM-based adjudication), and the paper's contribution is therefore more empirical and methodological than algorithmic. For TMLR, which values sound empirical contributions but also expects lasting scientific insight. Strengthening the novelty discussion and providing more rigorous validation would significantly improve the manuscript.

**Audience:**

Yes

**Audience Explanation:**

The findings are likely to be of interest to researchers working on data annotation, dataset construction, evaluation methodologies, sentiment analysis, information extraction, and LLM-assisted NLP workflows. As the community increasingly relies on large language models for synthetic data generation and annotation assistance, understanding the strengths and limitations of LLM-based annotators is an important practical question.

The paper contributes useful empirical evidence regarding where current LLMs succeed and fail when performing structured annotation tasks. In particular, the observation that span-level annotation quality remains relatively strong while relational annotation quality deteriorates substantially provides actionable insight for researchers considering automated dataset construction pipelines.

Although the work is application-oriented and does not introduce a fundamentally new machine learning algorithm, its findings contribute to an emerging literature on LLMs as annotation tools and are therefore likely to be relevant to a meaningful subset of the TMLR audience.

**Broader Impact Concerns:**

The work does not raise major ethical concerns beyond those already associated with automated annotation systems. However, the paper should explicitly discuss the risk of using LLM-generated annotations as substitutes for human annotations in subjective labeling tasks. The results indicate that LLMs may systematically distort relational structures while maintaining apparently strong span-level performance. Such errors could propagate into downstream datasets and models if not carefully monitored. A broader discussion of quality control mechanisms and human oversight would strengthen the manuscript.

**Claims And Evidence:**

Yes

**Claims Explanation:**

The paper generally supports its central claims with empirical evidence. The experimental evaluation covers multiple datasets, model scales, and annotation tasks, and the reported results consistently support the authors' primary conclusion that LLMs are substantially better at identifying individual sentiment-related spans than at reconstructing the relational structures that connect them.

The analysis goes beyond standard end-task F1 scores by decomposing performance into element-level and pairwise metrics. This decomposition is particularly valuable because it directly substantiates the paper's main argument regarding the distinction between span extraction and relational reasoning. The inclusion of inter-annotator agreement analyses further strengthens the study by providing an independent perspective on annotation consistency.

That said, some aspects of the experimental design could be strengthened. The adjudication model is optimized using ground-truth labels derived from the same benchmark ecosystem used for evaluation, which may inflate measured gains. Furthermore, statistical significance testing is not reported, making it difficult to assess whether some of the observed improvements are robust. The study would also benefit from evaluation on datasets outside the SemEval-style review domain to demonstrate broader generalization.

Despite these limitations, the evidence presented is generally clear, coherent, and sufficient to support the paper's primary claims.

**Requested Changes:**

Critical for acceptance:

1. Provide a more rigorous discussion of potential overfitting arising from prompt optimization and adjudication optimization on benchmark-specific development splits. The paper should clarify how much information from D1 and D2 influences the final adjudication system and whether alternative validation procedures were considered.

2. Include statistical significance testing or confidence intervals for the reported improvements, particularly for adjudication gains. Several reported improvements are relatively small, and it is unclear whether they are statistically meaningful.

3. Clarify the novelty of the adjudication framework relative to existing ensemble, self-consistency, and multi-agent aggregation methods. The current presentation may overstate novelty given the growing literature on LLM-based aggregation and adjudication.

Important but non-critical:

4. Evaluate the pipeline on at least one non-review-domain dataset to assess whether the conclusions generalize beyond SemEval-style sentiment benchmarks.

5. Include a stronger discussion comparing DSPy-based optimization with manually engineered prompts to quantify the value of the declarative framework itself.

6. Expand the error analysis by categorizing relational failures into specific error types (e.g., incorrect aspect-opinion linking, incorrect category assignment, implicit aspect hallucination, sentiment inversion).

7. Discuss computational cost, latency, and annotation throughput relative to human annotation workflows.

---

> ### Author Response · Authors · 2026-07-03
>
> # Response to Reviewer
>
> We thank the reviewer for the careful assessment. We lead with new experiments, then respond briefly to the remaining points. Numbers follow the reviewer's *Requested Changes*.
>
> *Note: In-text citations here are already in the Bibliography of the submitted manuscript.*
>
> ## New experiments: majority-voting baseline and statistical significance (Requests 2, 3)
>
> Naive element-wise voting is inapplicable to tuple sets of varying cardinality with no canonical alignment (Uma et al., 2021), so we implemented two set-level polling analogues: **Poll-exact** (keep a tuple if ≥2 annotators agree under exact match) and **Poll-flex** (relaxed span matching). Paired bootstrap tests compare each method against every annotator ($^{\ast}$ better than all three, $^{+}$ better than two).
>
> **ASTE** — F1 ($\mathcal{A}dj$ / Poll-exact / Poll-flex):
>
> | Data  | Mini                                  | Small                                 | Med                                   |
> | ----- | ------------------------------------- | ------------------------------------- | ------------------------------------- |
> | lap14 | **45.69**$^{\ast}$ / 43.54$^{+}$ / 39.91$^{+}$ | 40.24 / **45.03**$^{+}$ / 42.82   | 46.81 / **47.77**$^{+}$ / 44.83       |
> | res14 | **67.64**$^{\ast}$ / 64.73$^{+}$ / 61.29$^{+}$ | 61.65 / **66.02**$^{\ast}$ / 63.84$^{+}$ | **69.27**$^{+}$ / 68.99$^{+}$ / 66.69$^{+}$ |
> | res15 | **58.18**$^{\ast}$ / 56.21$^{+}$ / 52.53$^{+}$ | 56.38 / **59.53**$^{+}$ / 57.03$^{+}$ | 62.51 / **64.04**$^{+}$ / 61.29       |
> | res16 | 67.89$^{+}$ / **70.91**$^{\ast}$ / 66.85$^{+}$ | 62.69$^{+}$ / 64.63$^{+}$ / **65.51**$^{+}$ | **67.14**$^{+}$ / 64.63 / 63.83   |
>
> **ACOS** — F1 ($\mathcal{A}dj$ / Poll-exact / Poll-flex):
>
> | Data | Mini                          | Small                                 | Med                                            |
> | ---- | ----------------------------- | ------------------------------------- | ---------------------------------------------- |
> | Lap  | **16.43**$^{\ast}$ / 10.33 / 9.21 | **11.01**$^{\ast}$ / 8.15 / 9.00        | **18.99**$^{\ast}$ / 12.06$^{\ast}$ / 11.93$^{\ast}$     |
> | Rest | **31.63**$^{+}$ / 25.54$^{+}$ / 26.19$^{+}$ | **29.98**$^{\ast}$ / 25.12$^{\ast}$ / 24.13$^{\ast}$ | **40.53**$^{\ast}$ / 27.28$^{+}$ / 23.81$^{+}$ |
>
> Two points follow. The gains are **statistically meaningful**: aggregation significantly beats individual annotators in nearly all settings. And adjudication is *not* reducible to voting: on the harder ACOS task it beats both polling baselines in every cell, while on ASTE the two are competitive, the regime where a discrete vote suffices. This mirrors the paper's span-vs-structure bifurcation.
>
> ## Remaining points
>
> **Overfitting on $D_1$/$D_2$ (Request 1).** The development splits are used *only* for DSPy/MIPROv2 prompt optimisation without touching the model weights. The optimiser selects 5–15 in-context exemplars from DEV split (noted in Limitations), so the exposed information is far smaller than in SFT; a controlled SFT comparison is outside the scope of a training-free pipeline. We will make this explicit.
>
> **Novelty vs. ensemble / self-consistency / multi-agent aggregation (Request 3).** Self-consistency achieves reliability through redundancy, repeated samples of the *same* model (Wang et al., 2023), whereas annotation requires diverse sources: distinct annotators (persons or models), which our pipeline supplies and Poll-exact/Poll-flex combine by a fixed-rule vote. The adjudicator sits above this, a *learned* synthesiser in the spirit of stacked generalization (Wolpert, 1992) that beats both polling baselines on relational structure (ACOS). These designs are complementary, and our findings on where LLM annotation of opinion is aligned to human-annotators (and to what extent) informs the execution layer of a multi-agent ecosystems. Our contributions are framed as empirical and methodological we can further clarify in Related Work.
>
> **DSPy vs. manually engineered prompts (Request 5).** A manual baseline is confounded by construction: it measures the prompt author's skill, not the prompting *method*, given LLMs' sensitivity to instance-level phrasing (Zhuo et al., 2024). The declarative optimiser removes exactly this confound, so the polling/adjudication comparisons isolate aggregation effects rather than prompt-crafting proficiency.
>
> **Out-of-domain evaluation, error typology, and cost (Requests 4, 6, 7).** The element-wise/pairwise tables in the paper already localise the dominant relational failures, opinion-target mislinking ($at$ & $op$) in ASTE and aspect-category assignment ($ac$) in ACOS; the revision will name these as error categories and add pipeline throughput/latency (Kwon et al., 2023). Cross-domain evaluation beyond SemEval review data is left as future work.
>
> ## Final draft
> If given the opportunity, all of the above will be incorporated into the final draft.

---

### Review · Reviewer_6pGo · 2026-06-20

**Summary Of Contributions:**

The paper studies the utility of an adjudicator LLM over and above the LLM annotators. It uncovers a performance bifurcation, i.e. both LLMs and adjudicator are good at identifying the elements of opinion analysis, but are not good at tying them together. However, the work suffers from serious confounds (train and eval on the same split). The evidence presented is not statistically reliable (no significance tests whatsoever). Moreover, the work ignores the simplest possible baseline and does not present the only possible output (the final adjudication prompt). The paper requires significant revision and rewrite for it to be acceptable for publication.

**Additional Comments:**

N/A

**Audience:**

Yes

**Audience Explanation:**

Yes, LLM as a judhe is of great interest to TMLR's audience and understanding its mecahnism is interesting too, particularly without manual prompt-tuning, which the paper seeks to address.

**Claims And Evidence:**

No

**Claims Explanation:**

1. No aggregation baseline: The work rejects majority voting as "inapplicable" to combinatorial outputs, but this is incorrect: one can easily design a majority vote in O(1) time per sentence, O(n) for the dataset. No majority-vote, union, or intersection is reported. Table 8 hints Adj often just tracks A1. So the adjudication gain can't be trusted and could be matched by trivial aggregation at zero LLM cost.
2. The adjudicator is optimized and evaluated on the same split (D2): Annotators are optimized on D1 and evaluated on D2, which is good. But the adjudicator is optimized on D2, and the paper never states the Adj rows come from a held-out test split. The small 1-3% F1 gains (one +12 outlier) could be because of this.
3. D2 is also used for model selection (ranking A1/A2/A3, picking k). Selection, meta-training, and reporting on the same split is optimistic on every axis.
4. No statistical reliability: Single run, no seeds, no CIs, no significance tests. Core gains are 1–3 F1, well inside the above confounds' variance, so "adjudication consistently mitigates errors" is unestablished.
5. The adjudicator is the same model as annotator A1: a model tends to agree with its own outputs, and indeed the adjudicator's F1 closely tracks A1's.
6. The learned artifact is not shown: this paper is majorly about the optimised adjudication prompt, yet it is never shown. How does it differ from the annotator prompts? The paper only uses vague phrases like "resolve conflicts" and "synthesize", never describing "how".

Numerical errors
1. Result out of range: Laptop `op&s` reads "36.75" on a 0–1 α scale (probably ≈0.37).
2. Overstated threshold claim: The text says polarity F1 "frequently exceeding 90% F1." Only some cells reach ≥90, the rest are 83–90. "Approaching 90%" would be accurate.

**Requested Changes:**

Major
1. Add a non-LLM aggregation baseline: element-wise majority vote (with span matching), set union, and set intersection over the three annotations, reported in the same tables as the LLM adjudicator.
2. Have a test split completely unseen by the annotators and adjudicator at any point. Introduce a third partition so the adjudicator is never graded on its own training data, and report all final numbers on this held-out test split.
3. Do not use the same split for selection, meta-training, and reporting. Use one slice to rank A1/A2/A3 and pick k, a disjoint slice to generate redundant annotations and train the adjudicator, and the test split to report.
4. Add statistical tests. Run with multiple random seeds and report mean ± CIs, marking which Adj−A1 gains are significant (paired bootstrap or approximate randomization). This establishes whether the F1 gains (and the Small-tier reversal) are real rather than optimization noise.
5. Adjudicator should be different from the annotators. Run it with a model that is not A1 (e.g., a held-out or third-party model) to measure how much of the gain is genuine arbitration versus self-agreement.
6. Please release the optimized prompt (the learned artifact). Release the final MIPROv2-optimized prompts for both the annotators and the adjudicator. Showing how the adjudication prompt differs from the annotation prompt is what lets a reader understand the mechanis.

Minor
1. exact-match F1 is used for the main tables while the paper itself argues these tasks are subjective and that exact match over-penalizes boundary differences. No partial match metric is reported alongside.
2. Describe the adjudicator's actual decision procedure (does it select a whole annotator's set or compose tuples element-by-element, and how are opinions matched across annotators), and include a behavior breakdown showing how often Adj copies a candidate's element versus generates a span none of A1/A2/A3 proposed.

---

> ### Author Response · Authors · 2026-07-02
> **Additional Results and Clarifications**
>
> We thank the reviewer for a thorough, constructive review. As suggested, we performed additional majority voting baselines and significance tests. On Data Split & Leakage concerns: this was a writing issue, not a protocol issue. All reported numbers are computed on the original SemEval test splits, never used for prompt optimisation, annotator ranking, $k$ selection, or adjudicator training. “Evaluation on $\mathcal{D}_2$” refers to pipeline preparation (intermediate model/$k$ selection, akin to hyper-parameter search on a dev set). We will make data usage crystal clear when updating the draft.
>
> ## Majority Voting baseline and statistical reliability
>
> Raw element-wise majority voting was not directly applicable as annotators produce tuple sets of varying cardinality with no canonical alignment. As advised, two set-level polling aggregators were implemented over the three LLM annotators’ outputs: Poll exact (keep a tuple if ≥2 annotators propose it under exact match) and Poll flex (agreement under relaxed span matching).
>
> We also ran paired bootstrap significance tests for all main results, comparing each aggregation method against each individual LLM annotator (Does aggregation help?). Markers: $^*$ significantly better than all three annotators, $^+$ better than two.
>
> **ASTE Results** (F1 for: $\mathcal{A}dj$ / Poll exact / Poll flex):
>
>
> | Data  | Mini                                  | Small                                 | Med                                   |
> | ----- | ------------------------------------- | ------------------------------------- | ------------------------------------- |
> | lap14 | **45.69**$^{\ast}$ / 43.54$^{+}$ / 39.91$^{+}$ | 40.24 / **45.03**$^{+}$ / 42.82   | 46.81 / **47.77**$^{+}$ / 44.83       |
> | res14 | **67.64**$^{\ast}$ / 64.73$^{+}$ / 61.29$^{+}$ | 61.65 / **66.02**$^{\ast}$ / 63.84$^{+}$ | **69.27**$^{+}$ / 68.99$^{+}$ / 66.69$^{+}$ |
> | res15 | **58.18**$^{\ast}$ / 56.21$^{+}$ / 52.53$^{+}$ | 56.38 / **59.53**$^{+}$ / 57.03$^{+}$ | 62.51 / **64.04**$^{+}$ / 61.29       |
> | res16 | 67.89$^{+}$ / **70.91**$^{\ast}$ / 66.85$^{+}$ | 62.69$^{+}$ / 64.63$^{+}$ / **65.51**$^{+}$ | **67.14**$^{+}$ / 64.63 / 63.83   |
>
> **ACOS Results** (F1 for $\mathcal{A}dj$ / Poll exact / Poll flex):
>
> | Data | Mini                         | Small                           | Med                                   |
> | ---- | ---------------------------- | ------------------------------- | ------------------------------------- |
> | Lap  | **16.43**$^*$ / 10.33 / 9.21 | **11.01**$^*$ / 8.15 / 9.00     | **18.99**$^\ast$ / 12.06$^\ast$ / 11.93$^\ast$ |
> | Rest | **31.63**$^+$ / 25.54$^+$ / 26.19$^+$| **29.98**$^\ast$ / 25.12$^\ast$ / 24.13 $^\ast$ | **40.53**$^\ast$ / 27.28$^+$ / 23.81$^+$ |
>
> On ASTE, strict polling (as advised) is a strong baseline our paper should not have omitted as it is competitive or better in several settings, winning the entire Small tier. The adjudicator's advantage emerges on ACOS (better than both polls in all 6 settings, significantly better than all annotators in 5/6), where relational complexity grows and exact inter-annotator agreement collapses. The revision plans to add both baselines to the main tables, the polling algorithm to the appendix, and a clearer description of data usage.
> ## Adjudicator Selection Rationale ($\mathcal{A}dj =\mathcal{A}_1$)
> Appointing the strongest annotator as adjudicator mirrors annotation practice, where a senior annotator adjudicates (Hovy & Lavid, 2010;Klie et al., 2024). The bootstrap tests address the self-agreement concern: a highly self-preferential model would be statistically indistinguishable from $\mathcal{A}_1$, yet the adjudicator is significantly better than all three annotators in 8/18 settings ($^*$), and 5/6 on ACOS, hinting at adjudication exploiting $\mathcal{A}_2$/$\mathcal{A}_3$'s candidates. We agree a third-party adjudicator is a valuable control and will add it to the Limitations, anchored in the broader limitation of inheriting the human adjudication workflow.
> ## Releasing Optimised Prompts
> The revision will release all MIPROv2-optimised annotator and adjudicator prompts; we can also post the adjudication prompt in this thread (or repo link) if required.
> ## Minor & numerical
> - **m1:** partial-match metrics are **already reported** as element-wise/pairwise decompositions (Tables 5–6); we will cross-reference them where the subjectivity argument is made.
> - **m2:** the adjudicator performs a **joint extraction** conditioned on the text and the three candidate sets, outputting a full annotation set in one pass; Error analysis in Table 8  qualitatively analyses this behaviour. Additional analysis can be included if necessary.
> - **Numerical errors:** Typo in Table 7 will be fixed and the polarity claim will read "approaching, and in some settings exceeding, 90% F1."
>
> We thank the reviewer again for their helpful suggestions. If given the opportunity, we will include all of the above additions in the revised manuscript.

---

> > ### Comment · Reviewer_6pGo · 2026-07-18
> > **Response to Authors**
> >
> > I thank the authors for all the work and additional experiments.
> >
> > My remaining concerns center on the aggregation baseline.
> >
> > 1. Set-level polling is not element-wise majority voting
> >
> > The implemented baselines (Poll exact / Poll flex) vote at the granularity of the whole tuple: a tuple is retained only if ≥2 annotators propose that entire tuple. Element-wise majority voting operates at the granularity of the individual field, tuples are first aligned across annotators, then each field (aspect term, opinion span, polarity, and for ACOS, category) is voted on independently.
> >
> > The distinction is big. For example, if three annotators produce (a1,b1,c1), (a1,b1,c2), (a1,b2,c2). Then, element-wise voting yields (a1,b1,c2). This is because a1 wins 3–0, b1 wins 2–1, c2 wins 2–1. Under Poll exact, however, each of the three tuples is proposed by exactly one annotator, so none reaches the ≥2 threshold and the baseline returns empty. Set-level polling cannot synthesize the majority tuple, even when a clear per-field majority exists.
> >
> > This matters because exact whole-tuple agreement is extremely difficult as the number of fields increases. Requiring two or more annotators to match on all four ACOS fields is far more stringent than requiring per-field majorities. The reported numbers reflect this: Poll exact scores 43.5–70.9 F1 on ASTE (3 fields) but collapses to 8.2–27.3 on ACOS (4 fields). The single result favoring the adjudicator is therefore at least partly an artifact of the baseline's construction, rather than evidence of the method's superiority.
> >
> > I acknowledge that annotators produce tuple sets of varying cardinality. The same relaxed matching already used for Poll flex can be used here, take per-field majorities within each matched group, and apply a ≥2 inclusion vote for unmatched tuples. This is O(1) per sentence for a fixed annotator count and bounded opinions per sentence.
> >
> > To be clear, element-wise voting would not resolve every ACOS case. Where the three annotators split 1–1–1 on category (e.g. Table 8, Text 3: loc#gen / serv#gen / ship#gen, with the adjudicator recovering the gold rest#gen), no voting scheme can recover the correct label. This points to a different explanation for the ACOS gain, one the rebuttal does not offer: because the adjudicator receives the source text and performs a fresh joint extraction, it can emit labels no annotator proposed. That is re-extraction rather than aggregation. A control would settle it, run A1 once more on the same inputs without the three candidate sets. If that recovers most of the ACOS gain, the advantage comes from an additional extraction pass, not from adjudicating candidates.
> >
> > 2. The ASTE results do not support the paper's central claim
> >
> > From the rebuttal's own table, Poll exact outperforms the adjudicator in 7 of 12 ASTE settings, by margins up to +4.79 F1. By the authors' own significance counts (8/18 overall, 5/6 on ACOS), only 3 of 12 ASTE settings show the adjudicator significantly better than all three annotators.
> >
> > In summary, even with a weak baseline, the results undercut the paper's central claim that adjudication helps.